# First Isolation of a Herpesvirus (Family *Alloherpesviridae*) from Great Lakes Lake Sturgeon (*Acipenser fulvescens*)

**DOI:** 10.3390/ani12233230

**Published:** 2022-11-22

**Authors:** Amber E. Johnston, Megan A. Shavalier, Kim T. Scribner, Esteban Soto, Matt J. Griffin, Geoffrey C. Waldbieser, Bradley M. Richardson, Andrew D. Winters, Susan Yun, Edward A. Baker, Douglas L. Larson, Matti Kiupel, Thomas P. Loch

**Affiliations:** 1Aquatic Animal Health Laboratory, Aquatic Animal Disease Ecology Program, Michigan State University, East Lansing, MI 48824, USA; 2Department of Fisheries and Wildlife, College of Agriculture and Natural Resources, Michigan State University, East Lansing, MI 48824, USA; 3Department of Integrative Biology, College of Natural Science, Michigan State University, East Lansing, MI 48824, USA; 4Department of Epidemiology, School of Veterinary Medicine, University of California-Davis, Davis, CA 95616, USA; 5Department of Pathobiology and Population Medicine, College of Veterinary Medicine, Mississippi State University, Starkville, MS 39762, USA; 6United States Department of Agriculture, Agricultural Research Service, Starkville, MS 39762, USA; 7Department of Biochemistry, Microbiology, and Immunology, School of Medicine, Wayne State University, Detroit, MI 48202, USA; 8Michigan Department of Natural Resources, Lansing, MI 48909, USA; 9Department of Pathobiology and Diagnostic Investigation, College of Veterinary Medicine, Michigan State University, East Lansing, MI 48824, USA

**Keywords:** lake sturgeon, herpesvirus, great lakes, disease, genomics

## Abstract

**Simple Summary:**

Throughout the Great Lakes basin, infectious diseases likewise threaten wild and hatchery reared fishes, and often require management attention. The lake sturgeon (*Acipenser fulvescens*) is the only sturgeon species native to the Great Lakes, where it is the largest and longest living fish therein. Due to multiple known and unknown factors, current Great Lakes lake sturgeon populations are estimated to be <1% of historical abundances. One potential contributing factor about which little remains known, especially when compared to other Great Lakes fishes, is the impact of infectious diseases. To address this knowledge gap, a two-year disease surveillance study was undertaken, resulting in the detection and isolation of a herpesvirus from lesions observed on wild adult lake sturgeon in two Great Lakes watersheds (Erie and Huron). Genomic analyses revealed the recovered virus was most similar to, yet molecularly distinct from, a herpesvirus recently recovered from lake sturgeon in the Lake Michigan watershed (Wisconsin, USA). This newly described virus, proposed as Lake Sturgeon Herpesvirus 2, proved virulent to juvenile (<1 year old) lake sturgeon, whereby disease and mortality occurred in virus-exposed fish under laboratory conditions. Overall, results from this study highlight the potential threat this newly described virus poses to Great Lakes lake sturgeon conservation efforts.

**Abstract:**

The lake sturgeon (*Acipenser fulvescens*; LST) is the only native sturgeon species in the Great Lakes (GL), but due to multiple factors, their current populations are estimated to be <1% of historical abundances. Little is known about infectious diseases affecting GL-LST in hatchery and wild settings. Therefore, a two-year disease surveillance study was undertaken, resulting in the detection and first in vitro isolation of a herpesvirus from grossly apparent cutaneous lesions in wild adult LST inhabiting two GL watersheds (Erie and Huron). Histological and ultrastructural examination of lesions revealed proliferative epidermitis associated with herpesvirus-like virions. A virus with identical ultrastructural characteristics was recovered from cells inoculated with lesion tissues. Partial DNA polymerase gene sequencing placed the virus within the Family *Alloherpesviridae*, with high similarity to a lake sturgeon herpesvirus (LSHV) from Wisconsin, USA. Genomic comparisons revealed ~84% Average Nucleotide Identity between the two isolates, leading to the proposed classification of LSHV-1 (Wisconsin) and LSHV-2 (Michigan) for the two viruses. When naïve juvenile LST were immersion-exposed to LSHV-2, severe disease and ~33% mortality occurred, with virus re-isolated from representative skin lesions, fulfilling Rivers’ postulates. Results collectively show LSHV-2 is associated with epithelial changes in wild adult LST, disease and mortality in juvenile LST, and is a potential threat to GL-LST conservation.

## 1. Introduction

The lake sturgeon (*Acipenser fulvescens*; Family *Acipenseridae*) is the only indigenous sturgeon species in the Great Lakes Basin of North America, where it is also the largest and longest living fish [1]. Unique life history traits, such as delayed maturation, protracted spawning, periodicity and low natural recruitment, have placed lake sturgeon at increased risk for human-induced population declines [2]. Indeed, several anthropogenic factors (e.g., overharvesting, habitat degradation, dam construction) have been linked to declines in both numerical abundance and distribution range of Great Lakes lake sturgeon [3]. Despite substantial efforts to rehabilitate lake sturgeon, which include hatchery supplementation, their current abundance in the Great Lakes is estimated to be <1% of historical levels [1,3].

Despite substantial population declines, little is known about the infectious diseases of lake sturgeon compared to other sturgeon species (Order Acipenseriformes). This is a matter of concern given the propensity of disease to impede hatchery-based conservation and aquaculture productivity for sturgeon in North America [4,5] as well as worldwide [6,7]. In particular, viral diseases are important sources of mortality in a range of sturgeon species across North America, including white sturgeon (*A. transmontanus*), shortnose sturgeon (*A. brevirostrum*), pallid sturgeon (*Scaphirhynchus albus*), and shovelnose sturgeon (*S. platorhynchus*) [8,9,10,11,12].

Among the viruses associated with disease outbreaks in sturgeon, herpesviruses within the Family *Alloherpesviridae* have been identified as important pathogens of multiple sturgeon species [7,13,14]. The first report of a herpesvirus infection in sturgeon was from juvenile white sturgeon in California, USA, in 1990 [13]. This disease-causing virus, now known as Acipenserid herpesvirus-1 (AciHV-1; syn. white sturgeon herpesvirus 1), as well as Acipenserid herpesvirus-2 (AciHV-2; genus *Ictalurivirus*) [14], are both recognized as important sources of mortality in juvenile captive-reared white sturgeon in the western USA [13,14,15,16]. Subsequent studies have uncovered similar herpesviruses in Siberian sturgeon (*A. baerri*) in Russia [17,18], wild white sturgeon of the Columbia River (Oregon, USA) [19,20] and the Snake River (Idaho, USA) [19,20], and in cultured shortnose sturgeon from northeastern Canada [21]. Importantly, AciHVs have been isolated from the reproductive tissues of sexually mature and apparently healthy wild white sturgeon, raising the possibility of virus transmission from infected broodstock to offspring [14]. Despite this group of viruses having been detected in various North American sturgeon populations, an AciHV had never been detected in, or isolated from wild lake sturgeon. However, just prior to submission of the current manuscript, Walker et al. (2022) reported the detection of a novel alloherpesvirus in lake sturgeon in Wisconsin, USA, via a degenerate PCR assay and electron microscopy [22]. Although this virus, termed lake sturgeon herpesvirus, was found in association with skin lesions, in vitro isolation was not reported and its pathogenic potential was not determined.

Given the lack of reports of AciHV-1 and AciHV-2 in the Great Lakes basin in conjunction with their ability to cause sizeable mortality events in other sturgeon species, the Great Lakes Fishery Commission–Great Lakes Fish Health Committee designated these viruses (i.e., “white sturgeon herpesvirus”) as Emergency Fish Pathogens in the 2014 Model Program for Fish Management in the Great Lakes [23]. However, during a study investigating the infectious diseases of Great Lakes lake sturgeon conducted herein, a replicating agent with acipenserid herpesvirus-like characteristics was initially detected and subsequently isolated from grossly appreciable skin lesions in adult lake sturgeon from the Lake Huron and Lake Erie watersheds of the Great Lakes basin. As such, the objectives of this study following disease surveillance efforts were to initially characterize and identify the replicating agent and investigate the tissue changes associated with naturally occurring infections in adult lake sturgeon. Following the successful isolation of this alloherpesvirus, an additional goal was to characterize its capacity to cause disease and/or mortality in juvenile lake sturgeon via in vivo challenge experiments under controlled laboratory conditions.

## 2. Materials and Methods

### 2.1. Fish and Sampling

During May and June of 2019 (year one), adult lake sturgeon were sampled from two locations within the Great Lakes basin: (a) spawning or near-spawning phase adult lake sturgeon, captured via hand netting by Michigan State University personnel in the upper Black River (Lake Huron watershed, Cheboygan County, MI, USA); and (b) adult non-spawning lake sturgeon, captured via trotline by Michigan Department of Natural Resources (MDNR) personnel in the St. Clair River (Lake Erie watershed, St. Clair and Macomb County, MI, USA).

Upon capture, fish were manually and gently restrained while partially immersed in water for gross examination (<3 min) and non-lethal tissue collections. Biometric data (i.e., length, weight, girth) were recorded, after which, whole blood was collected by caudal venipuncture. A portion of whole blood was immediately transferred into a sterile heparinized tube (Heparin sodium salt, ThermoFisher Scientific, Waltham, MA, USA) for subsequent packed cell volume (PCV) measurement and 1 mL of whole blood was placed into a 1.5 mL tube for virological analyses, both tubes were placed in an ice-chilled cooler. If fish were ready to spawn, reproductive fluids and/or eggs were manually expressed, collected from the cloaca via syringe, saved in a sterile 5 mL centrifuge tube or Whirl Pak™ (VWR, Radnor, PA, USA), and immediately placed on ice. When skin lesions of concern were observed, a small amount of tissue was biopsied (<250 mg) using forceps and either a sterile scalpel or scissors. The biopsied tissue was divided and saved in RNAlater (Sigma-Aldrich, St. Louis, MO, USA) on ice for molecular analyses, in a 2 oz Whirl Pak™ on ice for virus isolation, and in some cases, the biopsied tissue was saved in 10% buffered formalin for histopathological analyses. During the year one field season, all samples for virus isolation were frozen at −20 °C after collection and prior to further analyses.

In May 2020 (year two), adult lake sturgeon were exclusively sampled from the Black River site due to the COVID-19 pandemic. All clinical examination and tissue sampling procedures were identical to year one, with the following exceptions: (a) within hours of collection, all reproductive fluid samples (750 µL per fish) were diluted in sterile viral growth media consisting of Earle’s salt-based minimal essential medium (EMEM; Invitrogen, Carlsbad, CA, USA) supplemented with 10% BD Bacto^TM^ tryptose phosphate broth (TPB; ThermoFisher Scientific, Waltham, MA, USA), 2% Fetal Bovine Serum (FBS; Gemini Bioproducts, Sacramento, CA, USA), penicillin (100 IU mL^−1^), streptomycin (100 µg mL^−1^; Invitrogen, Carlsbad, CA, USA), gentamicin sulfate (0.1 mg mL^−1^; Sigma-Aldrich, St. Louis, MO, USA), amphotericin B (2.5 µg mL^−1^; Sigma-Aldrich, St. Louis, MO, USA), and buffered with sodium bicarbonate (pH 7.4–7.6; Sigma-Aldrich, St. Louis, MO, USA) at a ratio of 1:4 (V:V), respectively; (b) whole blood samples were similarly diluted and collected directly into sterile viral growth media (1:10, V:V); (c) skin lesion tissues were collected directly into one mL of sterile viral growth media, not exceeding a dilution of 1:4 (W:V); and (d) all samples for virus isolation were maintained on ice and/or at 4 °C (i.e., unfrozen) prior to inoculation onto cells.

Differences in prevalence of grossly observable virus-suspect skin lesion (see below) by fish sex collected from the Black River in 2019 and 2020 were evaluated using Fisher’s Exact Test in Microsoft^®^ Excel^®^ (Version 2207 Build 16.0.15427.20182).

### 2.2. Hematology

Less than eight hours post-collection, heparinized blood from each fish was aliquoted into a micro-hematocrit tube. Whole blood was pelleted at 16,000 rpm in a CritSpin™ Hematocrit Centrifuge (Iris Sample Processing, Westwood, MA, USA) for 2 min and the packed cell volume (PCV) was measured following centrifugation [24].

### 2.3. Histopathology

Representative, formalin-fixed skin lesions with grossly observable thickened opaque foci were submitted to the Michigan State University Veterinary Diagnostic Laboratory, where they were paraffin embedded, sectioned (5 µm), and stained with hematoxylin and eosin (H&E) according to standard and routine methods [25]. Stained preparations were then examined via light microscopy for histopathological changes.

### 2.4. Virological Analysis via Cell Culture

#### 2.4.1. Cell Culture

Five cell lines were utilized for virus isolation: *Epithelioma papulosum cyprini* (EPC) [26], Chinook salmon embryo (CHSE-214) [27], white sturgeon skin (WSSK-1) [28], white sturgeon gonad (WSGO) [29], and a newly established white sturgeon × lake sturgeon hybrid spleen cell line (WSxLS) developed at the University of California Davis Aquatic Animal Health Laboratory following previously published protocols [30,31]. All cells were maintained in 75 cm^2^ cell culture flasks (Corning, NY, USA) at 25 °C (EPC) or 21 °C (CHSE, WSSK-1, WSxLS, WSGO). Growth media for EPC, WSSK-1, WSxLS, and WSGO cell lines were comprised of EMEM supplemented with 10% TPB, 10% FBS, penicillin (100 IU mL^−1^), streptomycin (100 µg mL^−1^), amphotericin B (2.5 µg mL^−1^), 2 mM L-Glutamine, and buffered with sodium bicarbonate (pH 7.4–7.6), CHSE cells were grown in Eagle’s MEM (with Earle’s salts, nonessential amino acids, and sodium pyruvate; ATCC, Manassas, VA, USA), nystatin (20 µg mL^−1^; Sigma-Aldrich, St. Louis, MO, USA), and L-glutamine, FBS, penicillin, and streptomycin as above. For virus isolation, cells were grown in 96-well, flat bottom plates (Corning, Corning, NY, USA) to 80–90% confluency (<48 h) in a medium as described above but modified to contain 2% FBS (WSSK, WSxLS; MEM-2-T) or 5% FBS (EPC, CHSE; MEM-5-T) and buffered with UltraPure Tris (Invitrogen, Carlsbad, CA, USA) to pH 7.4–7.6 [32].

#### 2.4.2. Sample Processing

For samples collected during year one, whole blood, reproductive fluids, and skin lesion tissues were thawed on ice and diluted (1:4 W:V) with EMEM supplemented with TPB, penicillin (100 IU mL^−1^), streptomycin (100 µg mL^−1^), gentamicin sulfate (0.1 mg mL^−1^), amphotericin B (2.5 µg mL^−1^), and buffered with UltraPure Tris (to pH 7.4–7.6; Invitrogen, Carlsbad, CA, USA). Samples were then homogenized and centrifuged (5000 RPM, 30 min, 4 °C) [33]. After centrifugation, whole blood sample supernatants were further diluted 1:5 with the sterile viral growth medium described above. For all tissues, the supernatant was incubated for 2 h at 15 °C or for 24 h at 4 °C in preparation for inoculation onto cells [32].

#### 2.4.3. Virus Screening and Isolation

Clarified supernatant was centrifuged just prior to inoculation (5000 RPM, 15 min, 4 °C) onto the cells. All samples from year one and two were inoculated onto EPC, CHSE, WSSK, and WSxLS, whereas a subset of samples from year one were inoculated onto WSGO. Use of the WSGO cell line was limited primarily due to cultivation challenges. After inoculation, all samples and cell lines were incubated at 15 °C. After a 14-day observation period, a second passage was performed on all samples, with no centrifugation step in between to allow for potential cell-to-cell virus transmission. Inoculated cell cultures were passed two to four more times for a total of 60 days on cells. If cytopathic effects (CPE) were noted, samples were passed into 25 cm^2^ culture flasks containing fresh cells (<48 h old). If CPE persisted, supernatant and cells were preserved at −80 °C in a solution of 20% FBS and 20% glycerol and/or frozen un-diluted for further analyses.

### 2.5. Electron Microscopy

For transmission electron microscopy, representative skin lesions that had been fixed in neutral-buffered, 10% formalin solution were trimmed into 2 mm pieces and postfixed in 1% osmium tetroxide in 0.1 M sodium phosphate buffer for 2 h. Tissues were serially dehydrated in acetone and embedded in Poly/Bed 812 resin (Polysciences Inc., Warrington, PA, USA) in flat molds. Sections were obtained with a Power Tome XL ultramicrotome (Boeckeler Instruments, Tucson, AZ, USA). To identify areas of interest, semithin (0.5-μm) sections were stained with epoxy tissue stain and examined under a light microscope. Ultrathin (70 nm) sections were then cut, mounted onto 200-mesh copper grids, stained with uranyl acetate and lead citrate, and examined under a 100 CXII transmission electron microscope (JEOL, Peabody, MA, USA). Similarly, WSxLS cells from cell culture samples (Section 3.4) exhibiting cytopathic effects (CPE) were harvested, centrifuged at 600 RPM (5 min, 15 °C) and the supernatant discarded. The cell pellet was then fixed in 10% buffered formalin and further processed as above.

### 2.6. Molecular and Gene Sequence Analyses

#### 2.6.1. Nucleic Acid Extraction

Representative skin lesion biopsies were processed for molecular analyses by thawing RNAlater (Sigma-Aldrich, St. Louis, MO, USA) preserved tissues at 4 °C, followed by centrifugation (5000 RPM, 10 min) to pellet the tissue and removal of stabilization buffer via micropipette. Following a single rinse with sterile phosphate-buffered saline (PBS), nucleic acid (DNA) extraction was performed using the DNeasy Blood and Tissue kit (Qiagen Inc., Germantown, MD, USA) according to the manufacturer’s protocol for animal tissue. The concentration of extracted DNA was quantified via a Qubit™ fluorometer using the double-stranded DNA broad range assay kit (dsDNA BR Assay Kit; Invitrogen, Carlsbad, CA, USA). Likewise, cell culture samples showing CPE, along with noninoculated negative control cells, were harvested for DNA extraction and subsequent PCR analyses. In brief, once samples were passed and CPE observed (5–14 days post passage onto fresh cells), DNA was extracted from samples using the DNeasy Blood and Tissue kit following the manufacturers protocol for cultured cells.

#### 2.6.2. Endpoint PCR Analyses

As the observed skin lesions were consistent with acipenserid herpesviral infections in other sturgeon species [13,14], degenerate PCR primers targeting a conserved region of the DNA polymerase gene of several alloherpesviruses were employed [34] using the cycling parameters of Kurobe et al. (2008) [18]. The HV (5′-CGG AAT TCT AGA YTT YGC NWS NYT NTA YCC-3′) and Cons lower (5′-CCC GAA TTC AGA TCT CNG TRT CNC CRT A-3′) primers amplify an ~500 bp region of the DNA polymerase gene of several alloherpesviruses, including AciHV-1 and AciHV-2 [18]. Each 25 µL reaction was comprised of 12.5 µL 2 × GoTaq Green Master Mix (ThermoFisher Scientific, Waltham, MA, USA), 5 µL of DNA template, and 40 pmol of each primer, with the remaining volume comprised of sterile, nuclease free water (Invitrogen, Carlsbad, CA, USA). Nuclease-free water, DNA extracted from apparently healthy and previously sampled adult lake sturgeon fin samples, as well as DNA extracted from uninoculated WSxLS cells served as template for negative control reactions, whereas NA extracts from previously confirmed AciHV-1 and AciHV-2 isolates were used as templates for positive control reactions. Each PCR was conducted in a Mastercycler Pro S 6325 Vapo protect PCR System (Eppendorf, Hamburg, Germany). PCR products were subsequently separated by gel electrophoresis in a 1.5% agarose gel infused with 0.0001% SYBR Safe DNA Gel Stain (ThermoFisher Scientific, Waltham, MA, USA; 1 mM) at 100 V for 30 min and then visualized via UV transillumination under a UVP DigiDoc-It Imaging System (UVP, Upland, CA, USA). A TrackIt™ 1 kb plus DNA ladder (Invitrogen, Carlsbad, CA, USA) was utilized for reference of amplicon size.

#### 2.6.3. Gene Sequence Analysis

Representative cell culture and skin lesion samples yielding amplicons of ~ 500 bp of the DNA polymerase gene target were purified using the QIAquick Spin Kit (Qiagen Inc., Germantown, MD, USA) and submitted to the Genomics Technology Support Facility (Michigan State University, East Lansing, MI, USA) for bidirectional Sanger sequencing using the HV and Cons primers of Hanson et al. (2006) [34]. Additionally, a subset of amplicons from PCR-positive cell culture and skin lesion samples were purified as above and subsequently cloned using the TOPO-TA cloning kit with TOP-10 *Escherichia coli* chemically competent cells (Qiagen Inc., Germantown, MD, USA) following the manufacturer protocols. Up to six resultant clones from each sample were screened for the target amplicon size via PCR analysis using primers M 13 Forward-20 (5′-GTA AAA CGA CGG CCA G-3′) and M 13 Reverse (5′-CAG GAA ACA GCT ATG AC-3′) and the following cycling parameters: 94 °C for 10 min, 25 cycles of 94 °C for 30 s, 55 °C for 30 s, and 72 °C for 45 s, and a final extension at 72 °C for 10 min. PCR products were visualized via gel electrophoresis as described above. Cloning-derived amplicons (*n* = 3–6 per sample) were purified, quantified, and submitted for sequencing as described above. Generated chromatograms were analyzed for quality and trimmed using Chromatogram Explorer (version 5.0.2.3). Contigs were assembled using the BioEdit sequence alignment editor (version 7.2.5) [35] via the Contig Assembly Program.

#### 2.6.4. Nanopore Sequencing and Genomic Analyses

Genomic DNA was prepared using the Puregene DNA kit (Qiagen Inc., Germantown, MD, USA) following the manufacturer’s protocol for cultured cells. Preparations from lake sturgeon herpesvirus isolate 200413-11TC, derived from a lesion sampled in an adult lake sturgeon in the Black River in year two (Section 3.4), were sequenced using a rapid barcoding kit (RBK-004, Oxford Nanopore Technologies, Oxford, UK) on a R9.4.1 flow cell. Raw Fastq sequences were filtered to a minimum read quality of 15 and a minimum length of 2kb using NanoFilt [36]. The filtered data, 220,725,293 bases in 57,034 reads, was assembled using Canu v2.0 [37]. The nanopore reads were re-mapped to the assemblies at a median 69-fold depth of coverage and consensus base calls were produced by Medaka v1.2.3 (https://github.com/nanoporetech/medaka [Accessed on 27 June 2022]) which produced two, non-overlapping contigs (119,069 and 87,627 bp).

The assembled contigs were annotated in Geneious Prime^®^ 2022.0.1 (Biomatters, Ltd., Auckland, New Zealand) based on GenBank Accession number (Acc. No.) OK485036 (Lake sturgeon herpesvirus strain Wolf River) [22] and regions corresponding to the 12 core alloherpesvirus ORFs [38] (DNA packaging terminase subunit 1 (Acc. No.: OK485036: ORF19), DNA polymerase catalytic subunit (Acc. No.: OK485036: ORF36), helicase-primase helicase subunit (Acc. No.: OK485036: ORF69), helicase-primase primase subunit (Acc. No.: OK485036: ORF26); major capsid protein (Acc. No.: OK485036: ORF53), capsid triplex protein 2 (Acc. No.: OK485036: ORF67), capsid maturation protease (Acc. No.: OK485036: ORF63), protein Allo37 (Acc. No.: OK485036: ORF 55), protein Allo54 (Acc. No.: OK485036: ORF33), protein Allo56 (Acc. No.: OK485036: ORF35), protein Allo60 (Acc. No.: OK485036: ORF28) and protein Allo64 (Acc. No.: OK485036: ORF25)), were extracted from the assembled contigs for analyses. Nucleotide sequences for the 12 core alloherpesvirus ORFs for lake sturgeon herpesvirus isolate 200413-11TC, have been deposited in GenBank (Acc. Nos. OP729406-OP729417). Initially, Blastn searches were performed for somewhat similar sequences of the NCBI non-redundant nucleotide (nr/nt) database for the 2100 bp coding sequence (CDS) of the DNA packaging terminase subunit 1 and the 4638 bp CDS of the DNA polymerase catalytic subunit of lake sturgeon herpesvirus isolate 200413-11TC.

Average nucleotide identity (ANI) estimations were made for the lake sturgeon herpesvirus Wolf River strain (Acc. No.: OK485036) and lake sturgeon herpesvirus isolate 200413-11TC using both best hits (one-way ANI) and reciprocal best hits (two-way ANI) as calculated by Goris et al. (2007) [39]. To account for the smaller viral genomes, combinations of minimum alignment lengths (125, 250, 500, 750 and 1000 bp) and minimum alignment identities (70%, 75%, 80%, 85%) were tested. Fragment options included a window size of 1000 bp and step size of 200 bp. To provide a frame of reference, similar comparisons were performed on two strains of Ictalurid herpesvirus 1 (channel catfish virus strain Auburn 1 [Acc. No.: NC_001493] and blue catfish alloherpesvirus strain S98-675 [Acc. No.: MK392382]) as well as comparisons between Cyprinid herpesvirus 1 strain NG-J1 (Acc. No.: JQ815363) and Cyprinid herpesvirus 3 strain GY-01 (Acc. No.: MK260013).

#### 2.6.5. Phylogenetic Analyses

Sanger generated sequences of the polymerase gene from discrete isolates were initially analyzed using the nucleotide Basic Local Alignment Search Tool (BLASTn) software from the National Center for Biotechnology Information (NCBI) nucleotide database. Reference sequences showing similarity were downloaded from NCBI, as were outgroups, and an alignment between study and reference sequences performed using ClustalW in MEGAX (Version 10.2.4) [40].

Similarly, MAFFT alignments were performed on nucleotide and translated amino acid sequences using the complete CDS of the 12 core alloherpesvirus ORFs from 17 alloherpesviruses for which complete or nearly complete sequences are available (Appendix A) [22]. ML trees of the translated amino acid sequences for the complete DNA packaging terminase subunit 1 and DNA polymerase catalytic subunit CDS, as well as the 12 core alloherpesvirus genes were generated using the Le_Gascuel_2008 model with Gamma distribution and invariable rates (LG + G + I) with 100 replicates [41]. Phylogenetic analyses of core alloherpesvirus genes were performed in MEGAX and resultant trees edited and refined in FigTree v1.4.2, Adobe Illustrator and Adobe Photoshop.

### 2.7. In Vivo Assessment of Isolated Herpesvirus Virulence to Juvenile Lake Sturgeon

#### 2.7.1. Origin of Fish for Virus Exposure Experiments

During May of 2020 and 2021, fish were spawned at the Black River Streamside Rearing Facility following previously described protocols and rearing practices [42,43,44,45,46]. In year one, fish were derived from four parental crosses, where parents were free of gross signs of disease. In year two, fish were derived from a single parental cross, where parents were free of gross signs of disease. Upon reaching eight weeks of age, fish were transported to the Michigan State University–University Research Containment Facility (MSU-URCF), where they were reared under quarantine conditions in a 680 L flow through tank supplied with ultraviolet-treated, oxygenated well water (11 °C ± 2 °C; ~9 L/min). Fish were fed frozen blood worms (Brine Shrimp Direct, Ogden UT, USA) daily (~2–5% body weight), and the tank was cleaned and siphoned daily to remove detritus/uneaten food. Prior to virus exposure, a subset of fish (*n* = 60) were screened via tissue culture for the presence of viruses, including AciHVs, and verified to be free from infection via standard laboratory protocols [32]. Briefly, fish were euthanized in tricaine methanesulfonate (MS-222; Syndel, Ferndale, WA, USA; 250 mg/L) buffered with sodium bicarbonate (500 mg/L), tissues collected, processed, and inoculated onto cells as detailed in Section 2.5. Additionally, tissues from representative 5 fish pools were screened with the Hanson et al. PCR assay [34] with subsequent cloning of amplicons as described in Section 2.6.3. No virus was detected via PCR in representative pools of tissue prior to conducting virus-exposure experiments.

#### 2.7.2. Preparation of Virus Inoculum for Exposure Experiments

Lake sturgeon herpesvirus isolate 200413-11TC was rapidly thawed from −80 °C at 25 °C and inoculated (25 µL) onto a 96 well plate of WSxLS cells prepared as described in Section 2.4.2 and Section 2.4.3 and incubated at 15 °C. Upon development of CPE, cells and supernatant were passed into 25 cm^2^ culture flasks and incubated further until CPE recurred. Virus infected cells/supernatant were then harvested and a Tissue Culture Infectious Dose_50_ (TCID_50_) performed by serially diluting virus 10-fold in sterile MEM-2-T (10^−1^–10^−7^) and inoculating all dilutions onto 96 well plates (eight replicate wells per dilution), which were observed for the development of CPE for 21 days, at which point wells with CPE were counted and the TCID_50_ calculated [47]. Virus was transferred on ice to MSU–URCF for exposure experiments.

#### 2.7.3. Pilot Exposure Experiments

To guide a replicated virus exposure study, multiple small pilot experiments were undertaken. Juvenile lake sturgeon (*n* = five/treatment; mean weight 1.74 g, Table 1) were exposed to Great Lakes lake sturgeon herpesvirus isolate 200413-11TC via intracelomic (IC) injection or bath immersion and after exposure, were maintained in flow-through (water temperature of 11 °C + 2 °C; ~1.5 L/min) or recirculating (water temperature of 22 °C ± 1 °C) 37.8 L aerated glass aquaria (Table 1). For IC injection exposure, fish were anesthetized with sodium bicarbonate-buffered (250 mg/L) MS-222 at a concentration of 125 mg/L and once sedate, injected with 25 or 50 µL (stock concentration: 1.53 × 10^2^–1.53 × 10^6^ TCID_50_/mL) of virus infected or negative control cell suspensions (Table 1) using sterile 1 mL syringes and 25-gauge needles. Following injection, fish were released back into their experimental tanks and monitored for recovery. For pilot bath immersion exposures, fish were immersed in a 125 mL suspension of virus (treatment group; 6.40 × 10^5^ TCID_50_/mL; Table 1) or cell growth media (negative control group) for 60 min at a concentration of one part virus suspension or negative control media to nine parts water (low dose) or one part virus suspension/media to four parts water (high dose). Following immersion exposure, fish and the virus or media suspensions were gently poured back into their respective 37.8 L flow-through glass aquaria.

Fish were monitored and cared for daily (Section 2.7.1). All dead fish were examined and multiple tissues (eye, fin, gill, kidney, spleen, heart, oral siphon, barbels, and/or skin lesions when present) aseptically collected, visceral organs and siphon/barbel samples pooled per fish, and processed as previously described for inoculation onto EPC, CHSE, WSSK, and WSxLS cell lines. Skin lesion samples were not pooled with other tissues and analyzed individually. Cell culture samples showing characteristic CPE observed on primary (i.e., first pass) inoculation were considered positive for viral infection. Terminally moribund fish and fish surviving until completion of the pilot experiments (i.e., 202–295 days post-challenge; Table 1) were euthanized via MS-222 overdose as above and analyzed similarly. All challenge experiments were conducted in accordance with the MSU-Institutional Animal Care and Use Committee (AUF: 201900057 and 202200033).

#### 2.7.4. Replicated Immersion Exposure Experiment

Guided by results from pilot experiments, a larger scale and replicated immersion exposure experiment was undertaken. For experimental inoculum, virus was cultured and quantified as described above. Fish (mean weight of 1.20 g; *n* = 8 fish per replicate tank, *n* = 6 replicates) were immersed in a 120 mL suspension of lake sturgeon herpesvirus isolate 200413-11TC at a concentration of one-part virus suspension to three parts tank water, equating to an exposure dose of 1.28 × 10^6^ TCID_50_/mL). Another six replicate tanks (*n* = 8 per replicate) were similarly mock challenged with one-part WSxLS cell/media suspension to three parts clean tank water with aeration. Following a 60 min immersion exposure, fish and the respective suspension were gently poured into their respective 37.8 L flow-through glass aquaria maintained at 11 °C ± 2 °C (flow rate of 1.5 L/min) with supplemental aeration. As detailed previously, dead fish were collected daily. Terminally moribund fish and fish surviving until the completion of the immersion challenge (112 days) were euthanized as described above and tissues (internal organs, siphon, barbel, fin, gill, lesions) collected for virus isolation and molecular analyses. Tissues for each fish were pooled as described above (visceral organs and siphon/barbel), however tissues from different fish were not pooled together. In brief, select cell culture supernatant samples showing CPE were analyzed using the primer set developed by Hanson et al. (2006) as described above [34], and a subset of samples were cloned and sequenced using methods described in Section 2.5, Section 2.6.1, Section 2.6.2 and Section 2.6.3.

## 3. Results

### 3.1. Gross Findings in Adult Great Lakes Lake Sturgeon

During year one, 137 spawning or near-spawning adult lake sturgeon (*n* = 111 males, *n* = 26 females) of varying lengths and weights (Table 2) were examined from the Black River. Among the examined fish, four (mean length, weight, and girth of 146.2 cm, 26.8 kg, and 63.8 cm, respectively) of 26 females (15.4%) and five (mean length, weight, and girth of 141.8 cm, 21.36 kg, and 53.4 cm, respectively) of 111 males (4.5%) from this location presented with focal (Figure 1a) to multifocal (Figure 1d) to coalescing (Figure 1b,c) semitranslucent to opaque, whitish, slightly raised plaque-like lesions on the fins, body, and/or head (Table 2). Likewise, among the 76 adult lake sturgeon examined from the St. Clair River in year one, one fish (1.3%; length of 120.8 cm, weight of 41.4 kg, and girth of 49.2 cm) had grossly similar skin lesions on the body and head (Figure 1d and Table 2). During year two, 79 adult spawning or near-spawning lake sturgeon (*n*= 54 males, *n*= 25 females) of varying lengths and weights (Table 2) were examined from the Black River. Among the examined fish, six (mean length, weight, and girth of 157.6 cm, 31.4 kg, and 60.25 cm) of 25 females (24%) and six (mean length, weight, and girth of 136.2 cm, 18.0 kg, and 50.25 cm) of 54 males (11.1%) presented with similar raised, opaque lesions on the trunk, fins, cranium, and opercula. Across all sites and sampling years, fish with skin lesions had a length, weight, and girth of 146.9 cm (Standard Deviation, SD: 13.42 cm), 24.7 kg (SD: 7.71 kg), and 55.25 cm (SD: 7.01 cm), compared to 136.3 cm (SD: 32.95 cm), 22.5 kg (SD: 21.10 kg), and 52.2 cm (SD: 14.783 cm) in fish without grossly appreciable skin lesions. When comparing the presence of skin lesions in males and females in 2019 using Fisher’s Exact Test, a one-tailed *p*-value of 2.42 × 10^−23^ was observed, indicating a significant difference in lesion prevalence. However, in 2020, a one-tailed *p*-value of 0.1371 was observed, indicating lesions prevalence was not significantly different by sex.

### 3.2. Hematology

Fish from all sites in year one and two had a mean PCV of 32.2%, with some variation by sampling site and year (Table 2). Overall, fish presenting with the skin lesions had a mean PCV of 33.9%, compared to a mean PCV of 32.6% in fish without skin lesions (Table 2). No clear trends in PCV were appreciable by site or year (Table 2); however, female lake sturgeon had lower mean PCVs in both year one and two at the Black River site when compared to males (Table 2). PCV measurements were comparable to reference intervals established by DiVincenti et al. (2013) and Cassle et al. (2013) [48,49].

### 3.3. Histopathology

Sections of fin, gills and the site of gross lesions were examined from five fish sampled from the Black River in year two. Fins and gills (lesions not collected for histopathology) were examined from two additional fish. All seven had skin abnormalities suspect for Acipenserid herpesviruses and were from the same site and year (Black River, year two). None of the fish had lesions in the gills. The microscopic changes in the gross lesion site were most severe in one fish of the five where lesion tissue was available, but overall similar in the other four fish. There was marked ballooning degeneration of the deeper portion of the stratum spinosum of the epidermis (Figure 2a). Affected cells were enlarged with lightly eosinophilic to clear cytoplasm and their nuclei were swollen with marginated chromatin (Figure 2b). Within the nuclei, there was often lightly eosinophilic, flocculent material forming central inclusions that displaced the chromatin. The more superficial portion of the epidermis had undergone necrosis and there was loss of nuclear detail, marked karyorrhexis and sloughing of epithelial cells. Multifocally there was transmural necrosis also affecting the superficial portion of the underlying lepidotrichia (Figure 2c). In the subepidermal portion of the connective tissue surrounding the lepidotrichia was loss of cellular detail and multiple foci of karyorrhexis. In the deeper portions of the connective tissue separating the mineralized spines was extensive neovascularization and proliferation of fibroblasts (granulation tissue) with dense infiltrates of small lymphocytes (Figure 2d). Multiple segments of the mineralized spines were partially lysed, had scalloped edges and were surrounded by proliferating fibroblasts or were completely lost in other areas (Figure 2e). There was common clustering of pigmented cells in the superficial connective tissue. In all submitted skin samples, there was marked epidermal hyperplasia with severe intercellular edema (spongiosis) in the stratum spinosum. Keratinocytes focally lost orientation, forming small clusters of cells, and mitotic figures were common. There were dense infiltrates of small lymphocytes in the lower portion of the stratum spinosum focally extending into the stratum basale (Figure 2f). Lymphocytes were commonly surrounded by a clear halo and surrounded individual necrotic keratinocytes. There was often clustering of pigmented cells within the superficial stratum spinosum of the more normal appearing epidermis forming a transition zone to the deeper portions of the epidermis that were infiltrated by lymphocytes.

### 3.4. Cell Culture

In year one, 119 whole blood, 91 reproductive fluid, and 8 skin lesion samples collected from the Black River and 57 whole blood, 3 reproductive fluid, and 1 skin lesion samples collected from the St. Clair River were inoculated onto the EPC, CHSE, WSSK-1, and WSxLS cell lines, with a subset also being inoculated onto WSGO. After a 60-day observation period, there was no observable CPE in any of the samples on any cell lines. In year two, 67 whole blood, 47 reproductive fluid, and 12 skin lesion samples collected from the Black River were inoculated onto EPC, CHSE, WSSK-1, and WSxLS. No CPE was detected after 60 days in any whole blood or reproductive fluid samples, nor in any of the skin lesion samples inoculated onto the WSSK, EPC, or CHSE cell lines. In contrast, CPE, in the form of large, multinucleated cells (i.e., syncytia), were observed on the WSxLS cell line (Figure 3) in 11 of the 12 skin lesions (91.6%), beginning as early as five days post inoculation. By two weeks post-inoculation, all cells had either detached, lysed, or formed syncytia. Subsequent passages resulted in identical syncytia formation and individual isolates were cultured and cryopreserved for further analyses.

### 3.5. Electron Microscopy

Transmission electron microscopy (TEM) revealed large numbers of herpesvirus-like particles (~85 nm) with electron-dense nuclei within the degenerating nuclei of affected cells (Figure 4a–d) from representative skin lesions, as well as WSxLS cells showing CPE.

### 3.6. PCR, Gene-Sequencing, and Phylogenetic Analyses

#### 3.6.1. Sanger Sequencing and Phylogenetic Analyses

Based on the gross similarity of the observed skin lesions to those associated with alloherpesviruses in other sturgeon species, the degenerate PCR assay of Hanson et al. (2006) targeting the DNA polymerase gene was used to test nine skin lesions from the Black (*n* = 8) and St. Clair (*n* = 1) Rivers in year one, and twelve skin lesions from the Black River in year two. Among them, five year-one skin lesions (*n* = 4 from the Black River and *n* = 1 from the St. Clair River) yielded ~500 bp amplicons, as did six year-two skin lesions from the Black River and six representative suspect virus isolates recovered on the WSxLS cell line. Direct Sanger sequencing of representative samples (Table 3) yielded sequences ranging from 468–529 bp and comparisons of quality trimmed (i.e., 458–477 bp) sequences using BLAST revealed highest similarity (ranging from 96.2–100%) to the newly detected lake sturgeon herpesvirus Wolf River strain (GenBank Accession Number: OK485036), followed by 79.0–82.0% similarity to an AciHV-1 DNA polymerase gene reference sequence (GenBank Accession Number: EF685904.1; Table 3). When representative amplicons from both sampling years and sites (3 from year one and 3 from year two) were cloned and subsequently sequenced (3–6 clones per sample), the generated sequences all were most similar to the same newly detected lake sturgeon herpesvirus Wolf River strain (GenBank Accession Number: OK485036; 98.7#x2013;100%; Table 3), followed by 81.0–82.0% similarity to an AciHV-1 DNA polymerase gene reference sequence (GenBank Accession Number: EF685904.1).

#### 3.6.2. Nanopore Sequencing and Genomic Analyses

The draft genome generated for lake sturgeon herpesvirus isolate 200413-11TC consists of two unique, non-overlapping contigs (119,069 and 87,627 bp). Blastn searches produced no direct matches to any alloherpesvirus isolates in GenBank (Appendix A) but did reveal a high degree of similarity (99.36%) across a 156-bp fragment of the DNA packaging terminase subunit I to lake sturgeon herpesvirus strains from the Chippewa (OL440177), Wolf (OL440173, OL440175) and Wisconsin (OL440170) Rivers in Wisconsin [22]. MAFFT alignments of the 4638 bp DNA polymerase catalytic subunit CDS from lake sturgeon herpesvirus isolate 200413-11TC with the Wolf River lake sturgeon herpesvirus (OK485036) and AciHV-1 from White Sturgeon (OK275729) revealed 92.6% nt (93.4% aa) and 79.8% nt (86.4% aa) similarity, respectively. Similarly, MAFFT alignments of the 2100 bp DNA packaging terminase subunit I CDS with other alloherpesviruses revealed isolate 200413-11TC was also divergent at this locus from the Wolf River lake sturgeon herpesvirus (86.8% nt; 92.8% aa) and AciHV-1 (85.7% nt; 92.0% aa). Similar levels of dissimilarity were observed for the remaining core alloherpesvirus ORFs (Table 4). Moreover, phylogenies based on concatenation of translated *ter1* and *pol* genes, as well as translations of the 12 core genes from 17 alloherpesviruses for which complete or near complete sequences were available (Appendix A [22,38,50,51,52,53,54,55,56,57,58,59,60,61,62]), were consistent with Blastn results, indicating lake sturgeon herpesvirus isolate 200413-11TC is most similar, yet distinct from lake sturgeon herpesvirus Wolf River (OK485036) and AciHV-1 from White Sturgeon (OK275729; Figure 5).

Across a series of alignment lengths and identities, the lake sturgeon herpesvirus Wolf River strain and the genomically characterized herpesvirus isolate recovered in the current study (Lake sturgeon herpesvirus isolate 200413-11TC) averaged 84.02% ANI (SD: 2.8074%; range 81.21–89.08%; Table 5), which is consistent with heterogeneity observed when comparisons were limited to the subset of 12 core genes. Comparably, across the same series of parameters, channel catfish virus and blue catfish alloherpesvirus averaged 94.10% (SD 0.53; range 93.38–94.82%) (Appendix A), while Cyprinid Herpesvirus 1 and Cyprinid Herpesvirus 3 were too divergent to make comparable analyses (Appendix A).

### 3.7. In Vivo Challenge Experiments

Following small-scale pilot challenges of juvenile lake sturgeon to multiple concentrations of the newly isolated lake sturgeon herpesvirus via multiple exposure routes (Table 1), gross disease signs, in the form of decreased reactivity to feed, lethargy, and peri-oral hemorrhage (Figure 6a), began to manifest at 3–31 days post-exposure (PE). Other signs of disease observed throughout the pilot studies included enophthalmia and focal dermal ulceration on the head and/or body of the fish (Figure 6b,c). Eventually, severe lethargy that progressed to fish lying in dorsal recumbency, severe ulceration of the caudal fin into the peduncle with associated hemorrhage (Figure 6d), and/or mortality began on day 8 PE and continued until day 84 PE (Table 1). Tissues from representative, grossly apparent skin lesions in both immersion and IC injection pilot experiments were inoculated onto WSxLS cells. After 7 days, syncytia were observed consistent with CPE from cells inoculated with lesion tissues collected from adult lake sturgeon. There was no evidence of CPE on CHSE, EPC, or WSSK, nor was virus recovered from other tissues sampled (i.e., oral siphon, barbel, eye, visceral organs).

Based in part on results from pilot experiments and experimental exposure studies with other acipenserid herpesviruses [13,14], an immersion route of exposure was selected for a larger scale, replicated challenge experiment. In the replicated immersion challenge, fish were exposed to lake sturgeon herpesvirus isolate 200413-11TC at a concentration of 1.28 × 10^6^ TCID_50_/mL, after which gross signs of disease were noted in virus-exposed fish beginning 10 days PE. Early signs of disease included moderate to severe lethargy when compared to negative controls across all replicates, and overall decreased reactivity to feed. Additionally, hemorrhaging around the oral siphon began 10 days PE (Figure 7a). At 15–16 days PE, more gross signs of disease began to manifest, including mild corneal opacity and mild-moderate erosion of the caudal fin. Within 48 h of caudal fin erosion onset, a subset of affected fish had near complete loss of the caudal fin and ulceration extending deep into the caudal peduncle musculature, often accompanied with moderate-severe hemorrhaging (Figure 7b,c). Disease signs continued to progress to marked emaciation, more widespread hemorrhaging around the oral siphon and caudal peduncle, and complete loss of the caudal fin and posterior portions of the caudal peduncle (Figure 7d). No gross signs of disease were noted in negative control, mock-exposed fish. Mortality in virus-exposed fish began 19 days PE and continued through 76 days PE, where all dead or terminally moribund and euthanized fish had grossly similar ulcerations of the caudal peduncle and resultant near complete loss of the caudal fin. Following conclusion of this experiment at 112 days PE, cumulative percent mortality (CPM) across all virus-exposed replicates reached a mean of 33.3% (Figure 8) compared to a mean CPM of 0.0% across all negative control, mock-exposed replicates.

Upon inoculation of collected tissues onto the WSxLS cell line, CPE (i.e., formation of syncytia) identical to what was observed in the herpesvirus infected adult lake sturgeon developed in representative skin lesion samples. Cell cultures inoculated with other tissues (i.e., siphon, barbel, visceral organs) produced no CPE. Subsequent PCR analyses of representative cells showing CPE using the degenerate primer set described above [34] yielded an ~500 bp amplicon; following cloning and Sanger sequencing, recovered virus isolates were confirmed to be the same lake sturgeon herpesvirus to which the fish were exposed, partially fulfilling Rivers’ postulates. There was no evidence of an active replicating agent in cells inoculated with tissues collected from survivors at the end of this study. Tissues from survivors were not analyzed by PCR.

## 4. Discussion

Herein, we describe the first in vitro isolation and virulence assessment of an alloherpesvirus from wild lake sturgeon in general, as well as the first detections of an alloherpesvirus in lake sturgeon inhabiting the Lake Huron and Lake Erie watersheds of the Great Lakes basin, North America. These findings are significant for several reasons. First, Great Lakes lake sturgeon populations are estimated to be <1% of historical abundances, and substantial resources are currently being allocated to the restoration of this iconic Great Lakes fish [1]. In this context, the present study not only demonstrated the presence of actively infectious virus was associated with skin lesions in wild adult lake sturgeon, but also that the newly isolated alloherpesvirus can induce severe disease and subsequent mortality in experimentally exposed juvenile lake sturgeon. These findings have both economic and environmental consequences, as these wild sturgeon serve as gamete sources for hatchery-based conservation and restoration efforts for these endangered animals in the Great Lakes region. Although this virus was not recovered from reproductive fluids in the current study, other alloherpesviruses from sturgeons (i.e., AciHV-1 and -2, white sturgeon) have been isolated from reproductive fluids of spawning adults with gross signs of disease [14], raising suspicions of possible transmission to and subsequent negative effects in offspring. Whether this newly detected virus has been or is leading to disease and/or mortality in wild and hatchery reared juvenile lake sturgeon remains unknown and warrants priority attention.

Second, other acipenserid herpesviruses (formerly known as white sturgeon herpesviruses) [13,14] are currently listed as emergency pathogens in the Great Lakes Fishery Commission-Great Lakes Fish Health Committee Fish Health Model Program [23]. This list, curated based on the potential threats certain pathogens pose to Great Lakes fishes, contains several significant fish pathogens believed to not yet be in the Great Lakes basin. As such, isolation of this virus is highly relevant to Great Lakes lake sturgeon management and may necessitate revision of the current list of emergency pathogens.

Third, alloherpesviruses have immunomodulatory effects in other fishes, leading to immunosuppression and increased potential for opportunistic infections [63,64]. For example, previous work found when common carp (*Cyprinus carpio*) were experimentally infected with Cyprinid herpesvirus 3, infection negatively affected the mucosal barrier of the skin and associated immune system activity [63]. Furthermore, infection with Salmonid herpesvirus 3 (EEDV) in lake trout (*Salvelinus namaycush*) induced skin ulcerations which were co-colonized with multiple fish-pathogenic bacteria, including *Flavobacterium psychrophilum* and *Aeromonas* spp. [64]. Similarly, Acipenserid herpesvirus 2 has also been linked to co-infections with *Streptococcus iniae* in white sturgeon (*A. transmontanus*) [16]. Collectively, literature supports the ability of alloherpesviruses to both negatively affect the immune system and subsequently provide opportunity for secondary or co-infections with other fish pathogens [16,63,64].

Lastly, herpesviruses are known to establish latent and chronic infections in hosts, where stressors, such as spawning or environmental factors could lead to recrudescent infections [64,65]. This was demonstrated during the resurgence of salmonid herpesvirus 3 (EEDV; *Alloherpesviridae*) in hatchery-reared lake trout (*S. namaycush*), where disease and mortality attributed to EEDV was thought to be induced by a combination of environmental stressors (e.g., temperature fluctuations and heavy rainfall) in addition to rearing densities [64]. Whether similar stressors and immunomodulatory effects are associated with infection emergence, status, or severity in the context of this newly isolated alloherpesvirus remains to be determined. Similarly, whether or not this alloherpesvirus establishes latency as other herpesviruses do is presently unknown [64]. Further, it is unclear if the targeted tissues and diagnostic tests deployed herein would have detected any latent infections. Consequently, investigating tissue tropism of the virus and documenting when and if it establishes a latent state would help optimize latent infection diagnostics [64].

Although the present study reports the first isolation of an alloherpesvirus from Great Lakes lake sturgeon, Walker et al. (2022) recently reported the molecular detection of an alloherpesvirus in four rivers in Wisconsin (US), including within the Lake Michigan watershed [22]. Following initial PCR screening for the alloherpesvirus DNA polymerase gene [34] and subsequent genome sequencing and assembly, Walker et al. (2022) determined the virus reported in WI lake sturgeon was an alloherpesvirus most similar to, yet distinct from AciHV-1. They proposed the name lake sturgeon herpesvirus to represent this newly characterized virus [22,34]. Interestingly, using the same degenerate primer set [34] and subsequent sequence analysis of partial DNA polymerase gene sequences, the virus detected in the present study was similar (~93% homology), yet not identical, to the lake sturgeon herpesvirus Wolf River strain [22,34]. Additional comparison of the genomes generated by Walker et al. (2022) and this study revealed high similarity across a small (i.e., 156 bp) portion of the DNA packaging terminase subunit I, yet shared <90% similarity across the entire CDS. Similarly, when compared across the other 10 core alloherpesvirus genes, the level of homology averaged 81.8% (range 68.9–86.3%). This was consistent with ANI comparisons across the available genomic data for the two lake sturgeon viruses, which revealed only 84% similarity using multiple alignment strategies (Table 5). Considering these results and acknowledging that strict criteria to delineate alloherpesvirus taxa are not fully established, we posit the genomically characterized virus detected in the current study is most similar to, yet distinct from, the lake sturgeon herpesvirus recently described by Walker et al. (2022) [22]. In fact, the %ANI between the lake sturgeon herpesvirus Wolf River strain detected by Walker et al. (2022) and the current study is ~10% lower than the %ANI for two catfish alloherpesviruses (i.e., channel catfish virus and blue catfish alloherpesvirus; Appendix A) that are believed to represent distinct taxa. Indeed, work done by Venugopalan et al. further supported the separation of channel catfish virus and blue catfish alloherpesvirus into separate taxa through host-specificity and serum neutralization studies [66]. Therefore, we propose the virus described in Walker et al. (2022) and the virus described herein be, respectively referred to as lake sturgeon herpesviruses 1 and 2 pending additional taxonomic investigations [22].

It is important to note that the geographic locales where these two viruses were recovered contain two genetically differentiated groups of lake sturgeon in the Great Lakes [67,68]. Furthermore, there is little mixing of individuals in Lake Michigan (Michigan vs. Wisconsin waters) [69], raising intriguing questions regarding the potential divergence of the two viruses since the potential lake sturgeon colonization of Lake Michigan from glacial refugia approximately 12,000 years ago [70].

Lake sturgeon herpesvirus 2 was detected in infected adult lake sturgeon residing in two (i.e., Huron and Erie) Great Lakes watersheds, where the virus was associated with grossly apparent skin lesions. Of note, these lesions were not only grossly and microscopically similar to those reported by Walker et al. (2022) [22], but also to white sturgeon infected with AciHV-2 [13]. Thus, it appears these viruses may have a similar disease manifestation despite representing different viral species and infecting different hosts. In the present study, the prevalence of these herpesvirus-linked skin lesions varied by sex in the Black River during both study years, whereby lesions were 2.2–3.4 times more prevalent in females than males. Infections with other fish pathogens (e.g., *A. salmonicida* and *Carnobacterium maltaromaticum*) in other fish species nearing and/or undergoing spawning have also been documented as more prevalent in females [71,72]. Potential factors behind the observed differences in virus-linked lesion prevalence by sex could involve the complex relationship between stress, sex, and the immune system [73]. Not only have female fish been documented to have higher cortisol levels during spawning that may dampen the immune response, but evidence also suggests female spawning fish have a less robust immune response compared to males [71,72,73,74]. Additionally, differences in the herpesvirus-linked skin lesion prevalence were apparent by sampling site, where prevalence was lower in the St. Clair River than the Black River in 2019 (1.1% vs. 6.6%; Table 2). Whether this relates to differences in spawning behavior, environmental factors and/or differences between the two sturgeon populations is unclear. It is noteworthy that in the Black River, nearly all fish were at the spawning phase, whereas fish in the St. Clair River were primarily pre-spawning phase, and spawning stress has been linked to emergence of active herpesvirus infections from latency and decreased immune competence [64,75,76].

Hematologic analyses revealed no significant differences in PCVs between fish with LSHV-associated lesions and those without. There is little published information regarding the relationships between hematologic parameters, such as packed cell volume, and alloherpesvirus infection. Overall, this study found no substantial differences in PCV between male and female fish from any site in any year, and all PCV averages fell within published sturgeon (*A. fulvescens* and *A. gueldenstaiedtii*) reference intervals (17–38% and 16–34%) [48,49].

To clarify the effects the newly recovered lake sturgeon herpesvirus may have on health and survival of juvenile lake sturgeon, in vivo laboratory challenge experiments were completed. Results indicate the newly isolated LSHV-2 is indeed capable of causing disease and subsequent mortality in juvenile lake sturgeon immersed in a virus-laden suspension under laboratory conditions. Likewise, these experiments revealed LSHV-2 was capable of replicating in naïve hosts and could be re-isolated from the resultant skin lesions, collectively fulfilling Rivers’ postulates [77]. Notably, the progression of grossly apparent disease signs was remarkably consistent across virus-exposed replicates, whereby fish developing gross disease signs showed lethargy and decreased reactivity to feed, which eventually progressed to complete loss of the caudal fin and deep ulceration into the caudal peduncle in every fish that died (*n* = 16). Somewhat surprisingly, the gross disease signs in LSHV-2-exposed fish showed uncanny similarity (e.g., ulcerative lesions) to those often associated with *Flavobacterium columnare*, the causative agent of columnaris disease [78,79]. However, tissues from caudal ulcerations were collected from representative fish in all replicate tanks and inoculated onto Hsu-Shotts medium using routinely employed methods [32,80] and no yellow pigmented bacteria, including *F. columnare*, were recovered (data not shown). Considering these findings, fishery managers and fish health professionals/clinicians should be aware that LSHV-2 appears to induce disease signs that could be readily confused with columnaris disease, potentially leading to delayed/misdiagnosis and/or ineffective treatments, especially without deployment of species-specific in vitro or specialized molecular techniques.

This study is the first to successfully isolate and propagate in vitro, a lake sturgeon herpesvirus. The CPE induced by the LSHV (i.e., syncytia formation) was highly similar to that noted in AciHV 1 and 2 [13,14,19]. Notably, propagation of this alloherpesvirus was only possible using a cell-line that included cells derived from the original host, the lake sturgeon, and is consistent with reports of strong host specificity often exhibited by alloherpesviruses [81]. Based on the present study’s results, it is tempting to speculate that LSHV-2 exhibits relatively narrow tissue tropism (e.g., the skin) given that it could only be recovered from the skin lesions of experimentally exposed fish; however, the sensitivity/limit of detection of the hybrid sturgeon cell-line that was utilized in this study is unknown and it is possible that a more sensitive assay might uncover virus presence in other tissues. Nevertheless, skin lesions clearly represent a priority tissue target for detecting/isolating LSHVs. In addition to requiring specialized cells and reagents, results from the current study affirm protocols established by the World Organisation for Animal Health (OIE), in that attempts to isolate an alloherpesvirus in vitro should avoid freezing of tissues [82]. Additionally, the use of a viral transport medium formulated based on recommendations from the American Type Culture Collection and American Fisheries Society-Fish Health Section [32,83] also seems to have enhanced successful virus recovery and may relate to the importance of antibiotics and fetal bovine serum for virus stability and replication. Based upon the results herein, when attempting to recover LSHVs in vitro, we recommend the following process be followed: (a) collect tissue and fluid samples directly into viral growth media (<1:4; W:V) detailed in Section 2.1; (b) maintain virus at 4 °C (i.e., unfrozen) and inoculate onto cells within five days of collection (less than 48 h is preferred); and (c) utilize cell lines derived, at least in part, from lake sturgeon.

## 5. Conclusions

This study describes the first isolation of an alloherpesvirus from wild adult lake sturgeon that concurrently exhibited gross cutaneous skin lesions. The recovered virus proved virulent to naïve juvenile Great Lakes lake sturgeon, whereby immersion exposure resulted in the development of disease and subsequent mortality and re-isolation of the virus from the skin lesions of affected hosts. Genetic and genomic characterization of the virus detected in this study revealed it was most similar but distinct from a very recently described LSHV (Walker at al. 2022) [22]. Based upon these results and until further taxonomic studies are undertaken, we propose the virus detected in Walker et al. and the virus isolated herein be referred to as lake sturgeon herpesvirus 1 and 2, respectively. Although much remains unknown about the effects LSHV-1 and 2 may have had or be having on LST populations, reports of the closely related Acipenserid herpesviruses being detected in reproductive fluids raises the potential for transgenerational transmission in lake sturgeon with unknown consequences for wild populations and hatchery conservation efforts alike. Research conducted in the present study is paving the way for ongoing and future research efforts seeking to arm those rearing lake sturgeon in the Great Lakes with efficacious management tools to reduce the negative health risks this virus poses to Great Lakes lake sturgeon populations and to their restoration.

## Figures and Tables

**Figure 1 animals-12-03230-f001:**
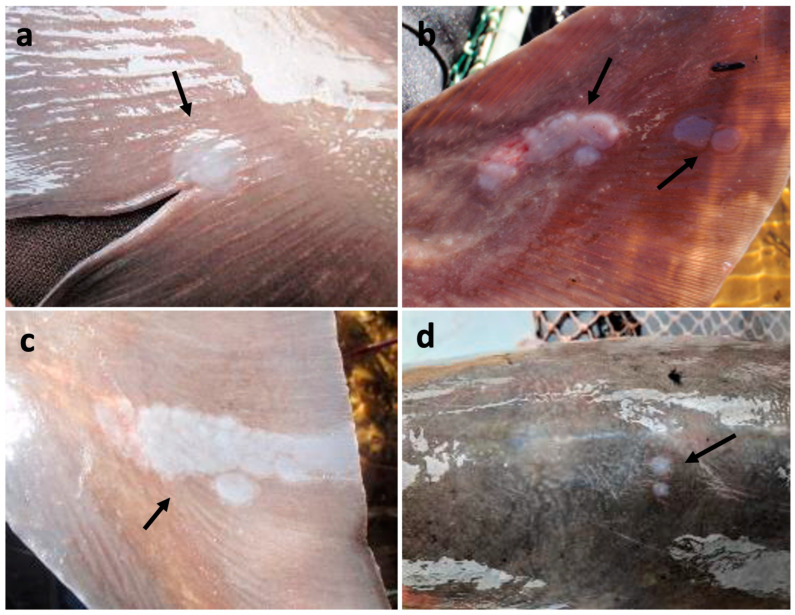
Semi-translucent to opaque, whitish, slightly raised plaque-like skin lesions observed on adult lake sturgeon in the Black (**a**,**c**) and St. Clair (**d**) Rivers indicated with black arrows. Lesion intensity varied from focal (**a**) to multifocal (**d**), to multifocal and coalescing (**b**,**c**) on the skin of the fins, head, and body.

**Figure 2 animals-12-03230-f002:**
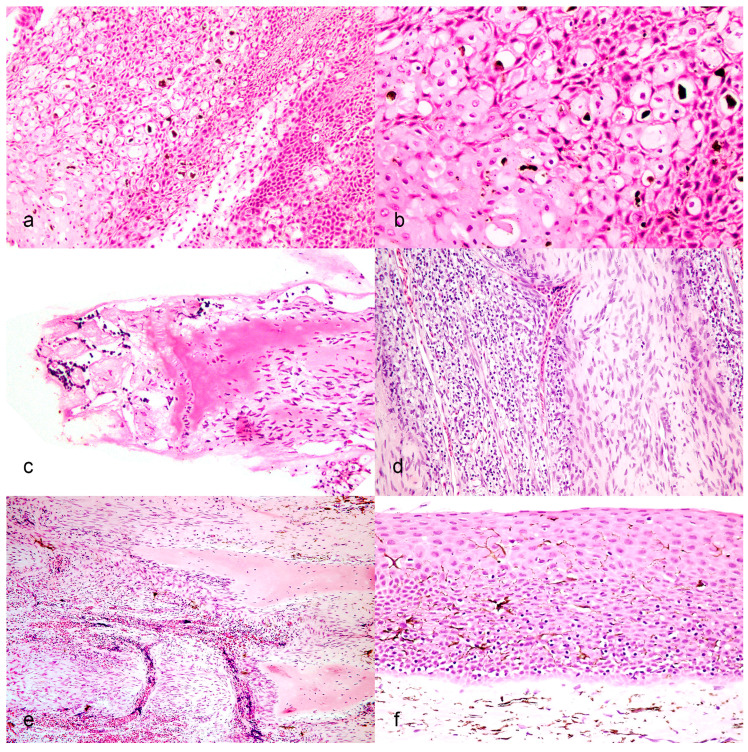
(**a**) Marked ballooning degeneration of the deeper portion of the stratum spinosum of the epidermis. (**b**) Affected cells are enlarged with lightly eosinophilic to clear cytoplasm and their nuclei are swollen with marginated chromatin. (**c**) Transmural necrosis affecting the superficial portion of the lepidotrichia. (**d**) Extensive granulation tissue with dense infiltrates of small lymphocytes separating mineralized spines. (**e**) Segments of mineralized spins were partially lysed, had scalloped edges and were surrounded by proliferating fibroblasts. (**f**) Dense infiltrates of small lymphocytes in the lower portion of the stratum spinosum focally extending into the stratum basale.

**Figure 3 animals-12-03230-f003:**
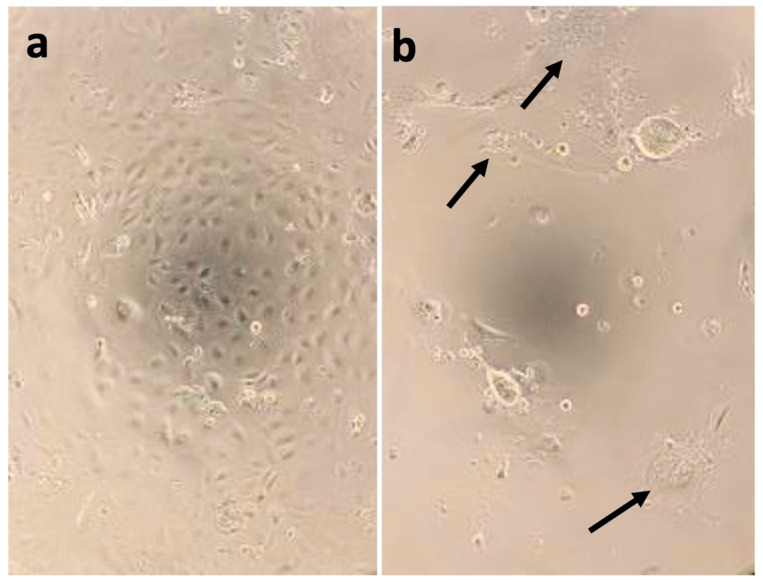
(**a**) A normal monolayer of white sturgeon × lake sturgeon (WSxLS) cells and (**b**) WSxLS cells showing cytopathic effects less than 14 days post-infection (15 °C) with tissue infected with virus. Arrows indicate large, multinucleated giant cells, or syncytia.

**Figure 4 animals-12-03230-f004:**
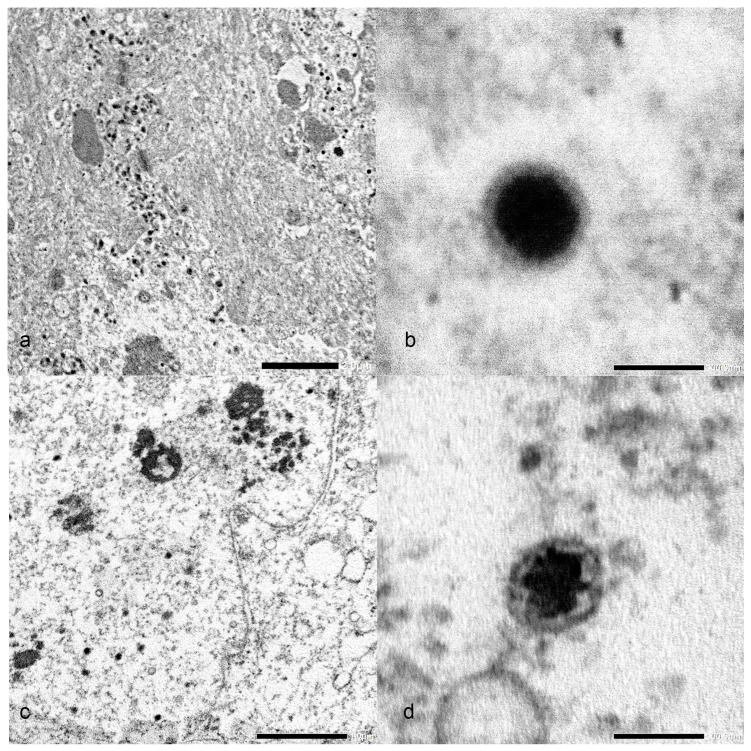
(**a**) Large numbers of herpesvirus particles (arrows) in degenerating nucleus of an epithelial cell in the stratum spinosum of the epidermis. bar = 2 µm (**b**) Higher magnification of mature 85 nm virion (C-capsid) from skin lesion. bar = 100 nm (**c**) Large numbers of herpesvirus particles (arrows) in degenerating nucleus of affected cell in cell culture. bar = 1 µm (**d**) Higher magnification of mature 85 nm virion from cell culture. bar = 100 nm.

**Figure 5 animals-12-03230-f005:**
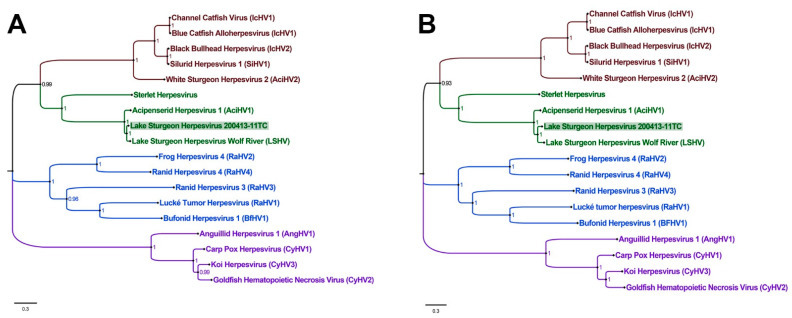
Dendrograms depicting the phylogenetic relationships between (**A**) translated and concatenated *ter1* and *pol* genes from 17 alloherpesviruses; and (**B**) translated and concatenated 12 core genes from 17 alloherpesviruses. In both dendrograms, lake sturgeon herpesvirus isolate 200413-11TC (this study; shaded) is well-supported as most similar to, yet distinct from, lake sturgeon herpesvirus Wolf River (OK485036) and AciHV-1 from White Sturgeon (OK275729).

**Figure 6 animals-12-03230-f006:**
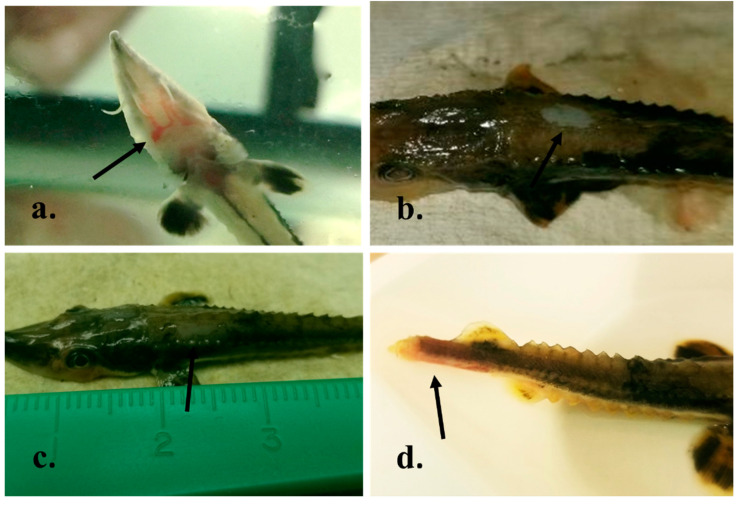
Gross disease signs observed in juvenile lake sturgeon (*Acipenser fulvescens*) exposed to the newly isolated lake sturgeon herpesvirus via intracolemic injection (**a**,**c**) or immersion bath (**b**,**d**); isolate 200413-11TC). (**a**) Hemorrhage surrounding the oral siphon extending linearly to the barbels indicated with the arrow. (**b**,**c**) Focal ulceration of the skin extending into the underlying musculature indicated with arrows. (**d**) Severe caudal fin erosion extending into the caudal peduncle (arrow), contiguous with marked pallor extending anteriorly to the trunk.

**Figure 7 animals-12-03230-f007:**
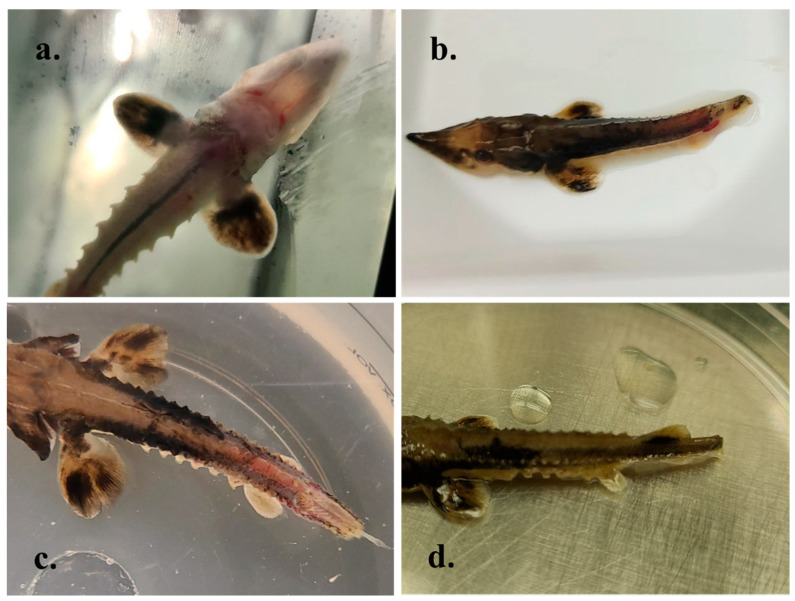
Gross disease signs in juvenile lake sturgeon (*Acipenser fulvescens*) exposed to the newly isolated lake sturgeon herpesvirus (isolate 200413-11TC) via immersion. (**a**) Hemorrhage surrounding the oral siphon extending linearly to the barbels 10 days post-exposure (PE). (**b**) Loss of the caudal fin and ulceration into the caudal peduncle (19 days PE). (**c**) Diffuse hemorrhage on the dorsal aspect of a lake sturgeon, along with deep ulceration of the caudal peduncle, 23 days PE. (**d**) Complete loss of the caudal fin and posterior portions of the caudal peduncle 34 days PE.

**Figure 8 animals-12-03230-f008:**
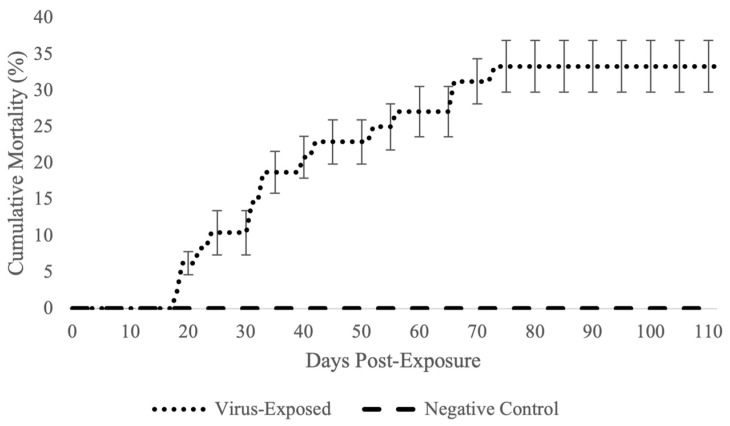
Cumulative percent mortality in juvenile Great Lakes lake sturgeon experimentally immersed in a suspension of the newly isolated lake sturgeon herpesvirus (1.28 × 10^6^ TCID_50_/mL). Error bars indicate standard error for every ten days post-infection.

**Table 1 animals-12-03230-t001:** Summary data for pilot in vivo experiments in juvenile Great Lakes lake sturgeon (*A. fulvescens*; *n* = 5/treatment/pilot experiment) that were exposed to the newly isolated Lake Sturgeon herpesvirus under multiple conditions and exposure routes. TCID_50_, Tissue culture infectious dose 50; IC, intracelomic; FT, flow-through; RC, Re-circulating.

							Cumulative Mortality (%)	
Fish Age (Weeks)	Mean Weight (g)	Tank Type	Water Temperature (°C)	Exposure Route	TCID_50_/mL	Duration (Days)	Exposed	Mock-Exposed
10.5 weeks	1.2	FT	11 ± 2	IC Injection	7.65	295	80%	20%
10.5 weeks	1.2	FT	11 ± 2	IC Injection	3.83	295	0%	20%
23.2 weeks	2.1	FT	11 ± 2	Immersion	6.40 × 10^5^	202	20%	0%
23.2 weeks	2.1	FT	11 ± 2	Immersion	3.20 × 10^5^	202	0%	0%
24 weeks	2.1	RC	22 ± 1	IC Injection	7.65 × 10^4^	208	20%	20%

**Table 2 animals-12-03230-t002:** Biometric, lesion prevalence, and packed cell volume (PCV, σ = Standard deviation) data collected from adult lake sturgeon in year one and two from 2 rivers within the Lake Huron (LH) and Lake Erie (LE) watersheds. Means and ranges are shown for biometric data.

Site	Lake Sturgeon Sampled	Total Length (cm)	Weight (kg)	Girth (cm)	Lesion Prevalence	PCV	PCV w/Skin Lesions
Black River Year 1 (LH)	137	148.6 (112–192)	24.5 (9.8–58.6)	57 (39–90)	6.6%	33.6% σ = 5.5	33.3% σ = 8.7
Females	26	154.1 (140–192)	40.3 (22.4–58.6)	73.1 (52–90)	15.4%	30.7% σ = 6.0	26.0% σ = 11.3
Males	111	147.55 (112–168)	20.8 (9.8–44.5)	53.2 (39–83)	4.5%	34.5% σ = 5.3	35.2% σ = 5.7
Black River Year 2 (LH)	79	150.45 (117–190)	26.62 (13.6–49.7)	55.05 (25–82)	15.2%	37.6% σ = 7.9	34.4% σ = 5.9
Females	25	160 (137–190)	33.3 (18.2–49.7)	63.8 (49–82)	24.0%	35.7% σ = 7.4	37.0% σ = 3.6
Males	54	145.9 (117–166)	23.0 (13.6–38.2)	51 (25–70)	11.1%	38.2% σ = 8.1	32.7% σ = 6.4
St. Clair River Year 1 (LE)	76	102.9 (27.2–176.1)	15.4 (0.6–29.9)	42.6 (9.5–78.1)	1.1%	25.6% σ = 7.6	38.0%

**Table 3 animals-12-03230-t003:** Accession numbers and origin of partial DNA polymerase gene sequences (length ranging from 458–477 bp).

Sample ID	Site	Year	Sequence Origin	Cloned?	Sequence Length (bp)	% Similarity to LSHV ^#^	% Similarity to LSHV ^±^	% Similarity to AciHV-1 *	Accession Number
C2019 LSC TIS 2	St. Clair River	2019	Skin Lesion	Yes	473	98.7	98.7	81.0	OP737376
2019 LSC TIS 2			Skin Lesion	No	473	97.0	96.8	81.0	OP737367
C2019 BLA TIS 6	Black River	2019	Skin Lesion	Yes	473	100	100	82.0	OP737377
2019 BLA TIS 6			Skin Lesion	No	473	99.6	99.6	81.6	OP737364
C2019 BLA TIS 16			Skin Lesion	Yes	473	98.9	98.9	81.2	OP737378
2019 BLA TIS 16			Skin Lesion	No	477	97.3	96.2	79.0	OP737365
2019 BLA TIS 41			Skin Lesion	No	474	99.0	99.0	81.2	OP737366
2020 BLA TC 2	Black River	2020	Cultured cells	No	458	98.5	98.3	80.4	OP737370
2020 BLA TC 7			Cultured cells	No	473	98.5	98.5	81.6	OP737371
2020 BLA TC 8			Cultured cells	No	473	100	100	82.0	OP737372
2020 BLA TIS 8			Skin Lesion		473	100	100	82.0	OP737368
2020 BLA TC 10			Cultured cells	No	473	99.4	99.4	81.6	OP737373
C2020 BLA TC 11			Cultured cells	Yes	473	99.8	99.8	82.0	OP737375
C2020 BLA TIS 11			Skin Lesion	Yes	473	99.4	99.4	81.8	OP737374
2020 BLA TC 25			Cultured Cells	No	473	99.8	99.8	81.8	OP737369
C2020 BLA TIS 25			Skin Lesion	Yes	473	99.8	99.8	81.8	OP737379

***** GenBank Accession Number: EF685904.1; ^#^ Lake Sturgeon Herpesvirus, GenBank Accession Number: OK485036; ^±^ Lake Sturgeon Herpesvirus sequence data from this study, GenBank Accession Number: OP729413.

**Table 4 animals-12-03230-t004:** Pairwise comparisons from MAFFT alignments of 12 core alloherpesvirus ORFs from lake sturgeon herpesvirus isolate 200413-11TC (Acc. Nos. OP729406-OP729417) to Lake Sturgeon Herpesvirus Wolf River and Acipenserid herpesvirus 1 UC-Davis [22].

	Lake Sturgeon Herpesvirus ^1^		Acipenserid Herpesvirus 1 ^2^	
	nt	aa	nt	aa
DNA polymerase catalytic subunit (*pol*; 4638 bp; OP729413)	92.6%	93.4%	79.8%	86.4%
DNA packaging terminase subunit 1 (*ter1*; 2100 bp; OP729416)	86.8%	92.8%	85.7%	92.0%
helicase-primase helicase subunit (*hel*; 1575 bp; OP729411)	83.1%	87.5%	81.6%	86.3%
helicase-primase primase subunit (*pri*; 2184 bp; OP729414)	78.5%	80.1%	77.3%	80.1%
major capsid protein (*mcp*; 3675 bp; OP729412)	86.3%	89.3%	77.3%	82.5%
capsid triplex protein 2 (*tri2*; 987 bp; OP729417)	83.9%	91.8%	79.0%	85.7%
capsid maturation protease (*pro*; 1659 bp; OP729415)	68.9%	66.2%	66.1%	65.8%
Allo37 protein (allo37; 1971 bp; OP729406)	84.6%	87.6%	77.7%	83.6%
Allo54 protein (allo54; 1857 bp; OP729407)	85.6%	91.4%	79.2%	84.5%
Allo56 protein (allo56; 3660 bp; OP729408)	80.9%	85.1%	79.3%	85.4%
Allo60 protein (allo60; 1053 bp; OP729409)	80.9%	85.1%	83.6%	86.6%
Allo64 protein (allo64; 1656 bp; OP729410)	80.1%	80.4%	78.2%	80.8%

^1^ Acc. No. OK485036. ^2^ Acc. Nos. OK275723, OK275724, OK275725, OK275726, OK275727, OK275728, OK275729, OK275730, OK275731, OK275732, OK275733, OK275734.

**Table 5 animals-12-03230-t005:** Average nucleotide identity estimations for Wolf River Lake Sturgeon Alloherpesvirus 1 (LSHV1; OK485036) and the Michigan State Strain of putative Lake Sturgeon Alloherpesvirus 2 (LSHV2).

LSAHV1 vs. LSAHV2				
Minimum Alignment	Minimum Identity	One-Way ANI 1	One-Way ANI 2	Two-Way ANI
125	70%	81.21% (SD: 6.62%), from 344 fragments.	81.36% (SD: 6.49%), from 342 fragments.	82.15% (SD: 6.96%), from 206 fragments.
125	75%	81.57% (SD: 6.41%), from 333 fragments.	81.67% (SD: 6.30%), from 333 fragments.	82.39% (SD: 6.80%), from 202 fragments.
125	80%	83.73% (SD: 5.53%), from 260 fragments.	83.64% (SD: 5.51%), from 264 fragments.	84.19% (SD: 6.13%), from 166 fragments.
125	85%	88.31% (SD: 5.05%), from 119 fragments.	88.17% (SD: 4.99%), from 123 fragments.	89.02% (SD: 5.64%), from 79 fragments.
250	70%	81.22% (SD: 6.61%), from 335 fragments.	81.27% (SD: 6.51%), from 338 fragments.	82.15% (SD: 6.96%), from 206 fragments.
250	75%	81.56% (SD: 6.42%), from 325 fragments.	81.61% (SD: 6.30%), from 328 fragments.	82.39% (SD: 6.80%), from 202 fragments.
250	80%	83.73% (SD: 5.55%), from 253 fragments.	83.61% (SD: 5.51%), from 259 fragments.	84.19% (SD: 6.13%), from 166 fragments.
250	85%	88.29% (SD: 5.10%), from 116 fragments.	88.12% (SD: 5.06%), from 119 fragments.	89.02% (SD: 5.64%), from 79 fragments.
500	70%	81.25% (SD: 6.65%), from 306 fragments.	81.31% (SD: 6.59%), from 305 fragments.	82.17% (SD: 6.97%), from 204 fragments.
500	75%	81.55% (SD: 6.48%), from 298 fragments.	81.65% (SD: 6.39%), from 296 fragments.	82.42% (SD: 6.81%), from 200 fragments.
500	80%	83.75% (SD: 5.64%), from 231 fragments.	83.64% (SD: 5.64%), from 234 fragments.	84.18% (SD: 6.15%), from 165 fragments.
500	85%	88.46% (SD: 5.23%), from 104 fragments.	88.34% (SD: 5.18%), from 106 fragments.	89.08% (SD: 5.65%), from 78 fragments.
750	70%	81.85% (SD: 6.59%), from 253 fragments.	81.99% (SD: 6.49%), from 255 fragments.	82.40% (SD: 6.85%), from 198 fragments.
750	75%	82.04% (SD: 6.46%), from 249 fragments.	82.19% (SD: 6.36%), from 251 fragments.	82.59% (SD: 6.72%), from 195 fragments.
750	80%	83.84% (SD: 5.80%), from 202 fragments.	83.72% (SD: 5.77%), from 210 fragments.	84.15% (SD: 6.16%), from 164 fragments.
750	85%	88.58% (SD: 5.47%), from 92 fragments.	88.38% (SD: 5.38%), from 97 fragments.	89.07% (SD: 5.69%), from 77 fragments.

## Data Availability

Data is contained within the article.

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
