# Peer review of "First Isolation of a Herpesvirus (Family Alloherpesviridae) from Great Lakes Lake Sturgeon (Acipenser fulvescens)"

_animals, 2022, doi:10.3390/ani12233230_

Round 1

Reviewer 1 Report (Previous Reviewer 2)

The authors might consider the following comments:  

1. Re response to item 12 in Reviewer 2 response table:  I could not identify which sub-section of the AFS Blue Book outlined the general procedure for creating a hybrid cell line.  Perhaps the authors could identify a more appropriate reference or provide more details.  

2. Re response to item 14 in Reviewer 2 response table: My comment here was specific to the naïve fish prior to their use in the virus exposure studies.  If the text provided in the authors' response (i.e.“In addition, tissues……(non-specificity)”) refers to PCR screening results from the 60 naïve fish, then the authors might consider adding this information to section 2.7.4.

3. Re response to item 22 in Reviewer 2 response table: The authors might consider adding “(PCR was not conducted.)” to the last line of section 3.7 to provide this information to the reader.

Author Response

Reviewer 2 Report (Previous Reviewer 1)

No further comments.

Author Response

This manuscript is a resubmission of an earlier submission. The following is a list of the peer review reports and author responses from that submission.

Round 1

Reviewer 1 Report

General comments

                The present paper describes the discovery of a novel herpesvirus in the Great Lakes in sturgeon showing clinical signs, named Lake Sturgeon Herpes Virus (LSHV). It seems to be distinct from previously reported herpesvirus in sturgeon species. Actually, there is a recent work (Walker et al., 2022) reporting a new sturgeon herpesvirus in Wisconsin. In that sense, the novelty factor of the manuscript reviewed here is low. The authors show virus-induced lesions in vitro and in vivo and compare the new virus sequence with that of other fish herpesviruses. Fulfilment of River´s postulates is demonstrated.

Due to the highly threatened status of lake sturgeon population, the surveillance of viral pathogens with capacity to cause mortality is a key factor in ensuring the species viability. Finally, it is not fully clear to me whether LSHV1 and LSHV2 are two different virus species or two local strains of the same virus. I´m guessing the experts on virus taxonomy will be deciding on that matter.

Specific issues

Figure 3 and caption. Please report day post-infection and temperature. I believe it would be an improvement to stain the cells to get a clearer image of the syncytia. Also, a higher magnification would help to clearly spot the nuclei.

Table 5. Are those sequences of the full viral genome?

Table 6 and 7 Perhaps these two tables could go into supplementary material.

Fig.6. I guess the route of infection was exposure to virus by bath immersion

Fig.9.  Same comment as for Fig.3. Also, in this figure, the addition of a picture of control (uninfected) spleen cells could help to better appreciate the cytopathic effect. This figure appears to have been taken at a larger magnification than figure 3. If that is the case, please indicate in the figure captions. Also for both figures 3 and 9: “multinucleated” syncytia, hhow many nuclei are we talking about? (it´s hard to tell from the pictures.

 One question: Herpesviruses often establish a latent infection in the host, with no apparent signs of disease. Could the LSHV be detected when in its latency state? (that is, from apparently healthy fish).

 A final comment: to my knowledge, the most cited viral pathogen in the Great Lakes appears to be the viral hemorrhagic septicemia virus (VHSV). Is it known if sturgeon can be a carrier of VHSV?

Reviewer 2 Report

A.       Summary. The authors isolated and characterized a herpesvirus (HV) from wild adult Lake Sturgeon (n=292) sampled during a 2 year field study conducted in the Lake Huron (Black River) and Lake Erie (St. Claire River) watersheds of the United States.  The herpesvirus diagnosis was based on evidence obtained from histopathology, virus isolation by cell culture on a WS x LS hybrid spleen cell line, electron microscopy, HV-specific molecular testing, phylogenetic analyses and next generation sequencing efforts.  River’s postulates were fulfilled following immersion exposure of juvenile Lake Sturgeon to the cultured virus.  Mortality (33%) was observed and exposed fish displayed the same lesions evident on the symptomatic wild adult sturgeon.  The challenge virus was re-isolated by cell culture from skin lesions (negative with other tissues) from dead or moribund fish collected throughout the in vivo study but not from survivors collected at the end of the study. 

B.       General comments.  The paper is well written and the topic falls within the journal’s scope.  The paper represents an important milestone that advances our understanding of sturgeon herpesviruses.  Some general comments are provided here.

1.       I suggest that the number of figures and tables be reduced by moving a portion of them to a supplementary file(s).  For example, consider presenting Tables 1, 3, 4, 6, 7 and Figures 6 & 9 as supplementary material.   

2.       The naming convention suggested by the authors (ie LSHV-1, LSHV-2) may lead to confusion in the context of the existing naming convention of AciHV-1 and AciHV-2.  In my mind, LSHV-2 would fall into the AciHV-2 clade and yet that would not be the case as LSHV-1 and LSHV-2 appear to be members of the AciHV-1 clade (see general comment 6 for more detail).  Perhaps they should reconsider their suggested naming strategy.  A possible alternative: if a hypothesis is made that the Lake Sturgeon viruses from the Great Lakes and the White Sturgeon viruses from California are both members of a AciHV-1 lineage (based on Fig. 5), then these Lake Sturgeon isolates would be AciHV-1 LSHV Wolf River OK485038 and AciHV-1 LSHV 200413-11TC.

3.       I recommend adding a table describing the Lake Sturgeon isolates from this study (i.e. those presented in Fig. 5) along with their Genbank Accession numbers. 

4.       A more fulsome description of the 2 contigs obtained from nanopore sequencing would be helpful.  Perhaps a figure would be the best way to illustrate the relative gene order. 

5.       Regarding contig description,  

a.       I made the assumption that the contigs were not overlapping.  Perhaps some statement to that effect could be included or add clarification that they in fact do overlap and the full genome length is 206,696 nucleotides.  If the contigs overlap then (e) below can be ignored.

b.       How does the arrangement of genes on the 2 contigs compare to the genome arrangement of the Wolf River isolate?  

c.        A description of the terminal repeats was not provided and would be helpful.

d.       What core alloherpesvirus genes were identified on the 2 contigs? 

e.       How did you confirm that the contigs/genome are from the same virus initially identified via the Hanson PCR? (For example, what was the result of comparing the DNA pol seq obtained from fragments amplified via the Hanson PCR test with DNA pol sequence from Nanopore sequencing?)

f.         What is the basis for concluding that both contigs are from the same virus (i.e. The DNA pol and terminase sub1 sequences were selected to conduct MAFFT alignments.  Were they selected because they are on different contigs and the BLAST analysis supported their origin from the same virus?). 

6.       Evolutionary relationships. The tree topology in Fig. 5 suggests that all the Lake Sturgeon isolates from this study cluster with the Wolf River isolate in a clade (represented by node labelled with a bootstrap value of 77) that may reflect host-specific adaptation of AciHV-1 isolates (ie White Sturgeon versus Lake Sturgeon).  In my mind there are 3 interesting evolutionary relationships to be explored with the genetic data: a) the relationship between the isolates from this study, b) the relationship between the Wolf River isolate and the isolate(s) from this study and c) their relationship to AciHV-1 White Sturgeon isolates.  I think that the authors can strengthen the evidence supporting their conclusion(s) regarding the relationship between the Lake Sturgeon isolates with their existing data.  I also think that they could explore more fully the broader relationship between the Lake Sturgeon isolates (Wolf River OK485038 and LSHV 200413-11TC) and the AciHV-1 White Sturgeon isolates.  The following comments are made in this context:

a.       The Lake Sturgeon isolates from this study: When the DNA pol sequences amplified by the Hanson et al PCR assay are aligned, does a pattern emerge in the nucleotide changes observed (between this study’s isolates? relative to the Wolf River isolate? as well as relative to the AciHV-1 White Sturgeon isolates?)?  If so, describe the pattern observed in section 3.6.  The authors could also consider including the % identity between the various Lake Sturgeon isolates from this study (ie Are they 100% identical across the sequenced fragment?).  The authors could consider presenting this finer resolution analysis to support their conclusion on lines 799-801 that the new isolates are genetically distinct from the Wolf River isolate. 

b.       Lake Sturgeon vs the AciHV-1 White Sturgeon isolates: Why are average nucleotide identity (ANI) estimations used when the more powerful phylogenetic-based methods would provide a better mechanism by which to explore the genetic relationship between their isolate and the Wolf River isolate (and more broadly the White Sturgeon isolates)?  I recommend performing phylogenetic analyses using a subset of the 12 core alloherpesvirus proteins/genes from the new virus genome (ex. full length DNA pol and terminase sub1).  If the authors want to keep the ANI results in the paper, then a comparison of the ANI results with the phylogenetic results could be included and in the discussion, the utility of the ANI approach for discerning virus evolution should be presented.  This investigation would produce a clearer and more definitive statement about the evolutionary relationships to support their conclusion that the isolates are different and provide further insight into their evolutionary relationship to existing sturgeon alloherpesviruses.

c.        An alternative: The authors could consider removing the genome sequence from this paper (submit in a separate paper).  In that case, I recommend completing 6a above and using the existing data from the Hanson et al PCR fragment to discuss the possible relationship between the Lake Sturgeon isolates from this study and the AciHV-1 White Sturgeon isolates.  The recommendations in 4, 5 and 6b would no longer apply to this paper.       

C.       Specific comments

1.       Abstract lines 39-40: Incomplete sentence - consider revising.

2.       Material and Methods line 166: No reference is provided for the hybrid WS x LS spleen cell line so I assume that it was developed as part of this study.  If this is the case, a description of how the cell line was generated should be included in the Materials and Methods section.

3.       Material and Methods lines 183-184: Homogenization – consider adding details about how the samples were homogenized.

4.       Materials and Methods lines 328-330: Were these 60 fish evaluated at any time for HV using the Hanson et al PCR test?  If so, then this information should be added as the PCR test would be more likely to detect the lower quantities of HV present in persistently infected fish.

5.       Materials and Methods lines 345-371:

a.       The authors might consider moving the pilot experiment description to the supplement.

b.       Note also that there are discrepancies between the text and the values presented in Table 1.  For example, line 349 the water temperature is 11+2C vs 10+1C in Table 1. Further, virus dose for IC exposure ranges from 1.53x10e2 to 1.53x10e6 TCID50/ml in Table 1 vs 1.53x10e2 to 3.2x10e5 TCID50/ml on line 353. 

c.        My calculations of virus dose administered in the IC virus exposure studies (based on Table 1) is 3.83 TCID50 for the 25 ul volume, 7.65 TCID50 for the 50 ul volume as well as 7.65x10e4 TCID50 for a later study using a 50 ul volume.  The first 2 doses seem very low.  Maybe I am misunderstanding something.  The authors should consider reporting the actual dose that was administered per fish by IC injection instead of their current way of presenting virus dose which requires the reader to calculate the final dose each fish received.   

6.       Materials and Methods lines 363-364: Was skin or skin lesion included?  Please indicate whether the tissues were pooled for virus isolation analyses.

7.       Materials and Methods lines 388-389: Please indicate whether the tissues were pooled for virus isolation analyses.

8.       Results lines 632-635: Consider revising sentence.  The result gets lost in the long sentence.

9.       Results line 635: Consider adding the tissues referenced here in brackets.

10.    Results lines 639-640: Were the survivors evaluated for HV using the Hanson et al PCR test?  If so, then this information should be added as the PCR test would be more likely to detect the lower quantities of HV present in persistently infected fish.

11.    Discussion line 719: Double-check whether reference 66 is appropriate.

12.    Discussion lines 776-793: The authors report that CPE was only observed when skin lesions (but not siphon, barbel, visceral organs, eye) were inoculated onto the hybrid cell line (e.g. lines 632-635). The authors may want to reiterate (in the discussion) the narrow tissue tropism displayed by the virus as an important consideration for sample collection to ensure the success of future cell culture-based diagnostic efforts. The impact of LSHV tissue tropism on molecular test results is less clear given that cell culture rather than PCR was used as the primary diagnostic method for the in vivo studies.